# Impact of spectacle use on academic performance among Vietnamese adolescents with reduced visual acuity and myopia: A school-based study

Hoang Thuy Linh Nguyen[1], Xuan Minh Tri Tran[1], Keiko Nakamura[2*], Kaoruko Seino[2,3], Yuri Tashiro[2], Ayano Miyashita[2], Tae Igarashi-Yokoi[4], Van Thang Vo[1], Kyoko Ohno-Matsui[4]

**1** Faculty of Public Health, Hue University of Medicine and Pharmacy, Hue University, Hue, Vietnam, **2** Department of Global Health Entrepreneurship, Institute of Science Tokyo (Science Tokyo), Tokyo, Japan, **3** National Institute of Public Health, Wako, Japan, **4** Department of Ophthalmology and Visual Science, Institute of Science Tokyo (Science Tokyo), Tokyo, Japan

* nakamura.ith@tmd.ac.jp

## Abstract

### Objectives

Untreated cataracts and uncorrected refractive errors are the leading causes of vision impairment worldwide. Timely correction of refractive errors can significantly improve visual function and daily life. This study examined the impact of spectacle use on the academic performance of schoolchildren with reduced visual acuity (VA) and myopia.

### Methods

This cross-sectional study included 647 students from five secondary schools in Hue City, Vietnam. Students underwent comprehensive eye examinations, including VA, autorefractometry, and axial length measurements, and a structured questionnaire was completed. The primary outcome was academic performance, based on grade point average (GPA), math, and literature scores. Generalized linear models were utilized to examine the association between spectacle use and students' academic performance.

### Results

The mean spherical equivalent was −0.92 ± 1.62 diopters, and the mean axial length was 23.32 ± 1.07 mm. The prevalence of reduced VA (uncorrected VA ≤ 7/10) and myopia (spherical equivalent ≤ −0.5 diopters) was 23.2% and 48.7%, respectively. Almost two-thirds of myopic students did not wear spectacles. Spectacle use was found to be significantly associated with an increased GPA (β = 0.462; 95% confidence interval = 0.108–0.816), math scores (β = 0.517; 95% confidence

**Data availability statement:** All relevant data are within the manuscript and its Supporting Information files.

**Funding:** This work is partly supported by Japanese Society for Promotion of Science Grant (17H02164). Japanese Society for Promotion of Science has no involvement in the study design; collection, analysis and interpretation of data; and writing the manuscript.

**Competing interests:** Authors declare that the research was conducted in the absence of any commercial or financial relationships that could be construed as a potential conflict of interest. This does not alter our adherence to PLOS ONE policies on sharing data and materials.

interval = 0.015–1.020), and literature scores (β = 0.438; 95% confidence interval = 0.074–0.802) among students with reduced VA, and with increased literature scores (β = 0.277; 95% confidence interval = 0.046–0.509) among students with myopia.

## Conclusions

Refractive errors, particularly myopia, are major health concerns among secondary school students in Vietnam. Despite the need for corrective spectacles, many students do not wear them, exacerbating their vision problems. This study highlights the benefits of spectacles in improving academic performance for students with reduced VA and myopia.

## Introduction

Myopia is one of the most common refractive errors globally. It presents a significant public health challenge, given its link to reduced visual acuity (VA) and the risk of vision-impairing complications [1,2]. Epidemiological studies estimate that approximately 10%–20% of school-aged children in low- and middle-income countries have myopia [3]. The Refractive Error Study in Children was conducted using the same sampling strategies, procedures, and definitions of myopia in different countries. The results showed varying prevalence of myopia across different populations, which were alarmingly higher in selected regions of Asia: 36.3% among 7–9-year-olds in Singapore and 42.4% among 13–17-year-olds in China. According to the Vietnam National Plan for Blindness and Prevention and Eye Care, uncorrected refractive error (36.4%) is a major cause of childhood blindness in Vietnam and is estimated to affect 15%–20% of Vietnamese children [4].

Research has shown a connection between parental myopia and myopia in children [5,6], but environmental factors also play a crucial role in the development of refractive errors, especially myopia [5]. Researchers have recently directed their focus to not only exploring risk factors but also understanding the potential roles of protective parameters [6–10]. Interestingly, controversial or unclear data still exist that explain potential or modifiable risk factors in developing countries or rural areas [11].

The World Health Organization's SPECS 2030 initiative aims for a world where everyone needing refractive error interventions has access to affordable, quality, and people-centered services [12]. The timely correction of refractive errors can improve productivity and functionality, which enhances participation in activities of daily living and visual function. Refractive surgical procedures, along with the use of spectacles and/or contact lenses, are commonly used to correct refractive errors, though they do not provide a complete solution for every individual. Globally, only 36% of people with distance vision impairment due to refractive errors have access to appropriate spectacles [13]. While spectacles cannot fully correct refractive errors, they offer a safe and affordable solution for managing these vision problems. Lower scores have been documented on a variety of motor and cognitive tests for children with uncorrected

refractive errors; however, reading, and subsequently the aforementioned scores, may improve when vision problems are corrected. To date, there have been no studies evaluating the effects of wearing spectacles on the activities of children who have refractive errors, specifically myopia, in Vietnam. Providing evidence of these effects may improve compliance with wearing spectacles and prompt families to seek treatment. This study, therefore, aimed to determine the effects of wearing spectacles on academic performance among secondary school students with reduced VA and myopia.

## Materials and methods

### Study population

This study was conducted in Hue City, in the central area of Thua Thien Hue Province in central Vietnam. Participants were selected from a baseline survey of an ongoing school-based cohort study ("Hue Healthy Adolescent Cohort Study") from 2018–2021. Hue City encompasses an area of 266 square kilometers and has a population of approximately 20,000 secondary school students. Participants for this study were selected using a multistage stratified cluster random sampling method to ensure a representative sample across different schools and class sizes. First, 5 secondary schools were randomly selected from among a total of 23 public secondary schools in Hue City. Second, depending on the size of the school, 4 or 5 sixth-grade classes (comprising 11-year-old students) were randomly chosen from each school. The initial sample size for the cross-sectional survey was 755 students. Of those invited, 647 students participated, resulting in a valid response rate of 85.7%.

### Procedures

All study participants underwent a comprehensive eye examination, which included the measurement of VA, autorefractometry, and assessment of axial length (AL). Participants with severe ocular diseases other than refractive error, such as cataracts, were excluded. Eligible participants underwent uncorrected visual acuity (UCVA) testing at a 5-meter distance, measured monocularly and binocularly, using a Landolt C chart under standard lighting conditions (approximately 100–300 lux). Autorefractometry was performed in a non-cycloplegic state using an autorefractometer (autorefractor ARK-1; Nidek, Aichi, Japan). AL was measured with an ultrasound biometer A-scan (ECHOScan US-4000; Nidek, Aichi, Japan) using the contact method. Prior to measuring the AL, corneal anesthesia was performed with one drop of 0.5% proparacaine hydrochloride (Alcaine 0.5%; Alcon Laboratories, Fort Worth, TX, USA) to minimize discomfort. All examinations were performed by trained ophthalmologists.

Each student also completed a structured questionnaire, developed at the Department of Global Health Entrepreneurship of Tokyo Medical and Dental University, based on the Global School-Based Student Health Survey [14] and Child Refractive Error Risk Factor Questionnaire [15], in a face-to-face interview by a trained team of staff. The questionnaire was first developed in English, then translated into Vietnamese and then back-translated to English for clarity and validity. The questions cover a broad range of activities of daily living, including study habits, time spent on electronic devices for pleasure such as playing games, and social media (Facebook, Instagram, YouTube) use. The Cronbach's alpha for variables concerning study time and recreational use of electronic devices was 0.6.

### Data analysis and definitions

UCVA was categorized into two groups: normal (VA > 7/10) and reduced VA (VA ≤ 7/10). UCVA was measured separately for each eye to determine the prevalence of reduced VA, with assessments conducted independently for the right and left eyes. The spherical equivalent (SE) was calculated as sphere + ½ cylinder. Myopia, emmetropia, and hyperopia were defined as ≤ −0.5 diopters (D) of SE, −0.5 D < SE < 0.5 D, and ≥ 0.5 D of SE, respectively. Only data from the right eye were used, as there was a high correlation in the SE between the right and left eyes ($r_{Spearson} = 0.87$, $P < 0.001$).

Wearing spectacles was defined based on whether students were observed wearing spectacles during the assessment. Total study time was defined as the sum of the time spent in extra classes and studying at school and home and

was recorded as "< 6 hours/day," "6–8 hours/day," or "> 8 hours/day". The daily time spent on electronic devices for pleasure was calculated using the following formula: $[(\textit{time spent on weekdays}) \times 5 + (\textit{time spent on weekends} \times 2)]/7$. These results were recorded and classified as "not at all," "< 1 hour," and "> 1 hour."

In the descriptive analyses, categorical variables are summarized using proportions and presented in tables. The chi-square test (categorical variables) and Mann–Whitney U test (continuous variables of non-normal distribution) were used to examine sex differences among participants' characteristics. Generalized linear models were used to investigate the relationship between wearing spectacles (independent variable) and academic performance (dependent variable), measured by continuous variables like grade point average (GPA), math scores, and literature scores. Model 1 was a univariate model including only the variable for wearing spectacles. Model 2 was adjusted for additional factors such as sex and study time. Model 3 was further adjusted for sex, study time, and recreational use of electronic devices. Data were analyzed using the Statistical Package for Social Sciences software (IBM SPSS Statistics for Windows, Version 22.0. Armonk, NY: IBM Corp), with statistical significance set at $P<0.05$. No imputation was performed for the missing data.

### Informed consent and ethical considerations

The protocol for this study was approved by the institutional review boards of the medical schools of Tokyo Medical and Dental University, Japan, and Hue University of Medicine and Pharmacy, Vietnam. Permission to recruit secondary school students was obtained from the Department of Education and Training in Thua Thien Hue Province, Vietnam. The investigative team prescheduled a meeting with the principals of the randomly selected schools and received consent before the onset of the study. All principals agreed to participate. All enrolled subjects agreed to participate after the purpose of the study was explained, and written informed consent was obtained from the parents/guardians and study subjects prior to their participation.

## Results

The participants' characteristics are presented in Table 1. A total of 647 (85.70%) of the 755 secondary school students (11-year-olds) invited participated in the study; 57.3% were girls, and 88% lived with both parents. The mean SE and AL of the right eye were −0.92 ± 1.62 D and 23.32 ± 1.07 mm, respectively. Statistically significant differences were observed between the mean SE and AL according to sex in both eyes ($P<0.05$). The prevalence of reduced VA in both eyes was 23.2%, with girls having a slightly higher prevalence than boys (13.3% vs. 9.9%, respectively; $P<0.05$).

**Table 1. Study participant's characteristics (n=647).**

| | Total (n=647) | Boys (n=339) | Girls (n=308) | P-value |
|---|---|---|---|---|
| Prevalence of reduced VA in both eyes[c], n (%) | 150 (23.2) | 64 (9.9) | 86 (13.3) | 0.006[a] |
| Prevalence of reduced VA in right eye, n (%) | 180 (27.8) | 76 (11.7) | 104 (16.1) | 0.001[a] |
| Prevalence of reduced VA in left eye, n (%) | 179 (27.7) | 76 (11.7) | 103 (15.9) | 0.002[a] |
| Spherical equivalent in right eye (D), mean (SD) | −0.92 (1.62) | −0.72 (1.45) | −1.14 (1.77) | <0.001[b] |
| Spherical equivalent in left eye (D), mean (SD) | −0.89 (1.72) | −0.72 (1.49) | −1.07 (1.94) | 0.010[b] |
| Axial length in right eye (mm), mean (SD) (n=645) | 23.32 (1.07) | 23.39 (0.96) | 23.13 (1.14) | <0.001[b] |
| Axial length in left eye (mm), mean (SD) (n=645) | 23.44 (0.97) | 23.59 (0.97) | 23.28 (0.94) | <0.001[b] |

VA: visual acuity; SD: standard deviation

[a]Pearson Chi-Square test; [b]t-test

Table 2 shows that the prevalence of spectacle use among the study population was 20.6%. Among students with reduced VA in both eyes, 25.3% did not wear spectacles, and 63.2% of students with myopia were also without spectacles. Moreover, Table 3 reveals that among students with reduced VA (n = 150), 94.0% had myopia, compared with 35.0% among those without reduced VA (n = 497).

Table 4 shows that wearing spectacles was significantly associated with improved academic performance among students with reduced VA, with increased GPA (β = 0.470; $P$ = 0.012), math scores (β = 0.534; $P$ = 0.040), and Vietnamese literature scores (β = 0.441; $P$ = 0.020) compared with those of students not wearing spectacles. After adjusting for sex, study time, and time spent on electronic devices for pleasure (model 3), wearing spectacles remained significantly associated with an increased GPA (β = 0.462; $P$ = 0.011), math score (β = 0.517; $P$ = 0.044), and Vietnamese literature score (β = 0.438; $P$ = 0.018).

Table 5 shows that for students with myopia, wearing spectacles led to higher GPAs (β = 0.237; $P$ = 0.045) and Vietnamese literature scores (β = 0.319; $P$ = 0.010) than for those without spectacles. These results were consistent with the multivariate linear regression (models 2 and 3), which showed that students with myopia who wore spectacles scored higher in Vietnamese literature than those without spectacles after adjusting for sex, study time, and time spent on electronic devices for leisure (β = 0.277; $P$ = 0.019).

## Discussion

This study aimed to determine the prevalence of myopia and reduced VA among secondary schoolchildren in Vietnam and assessed the effects of wearing spectacles on academic performance among children with reduced VA and myopia. Among the participants, 48.7% and 23.2% had myopia and poor vision, respectively. Although spectacles are considered to be a cost-effective way to treat myopia, the empirical evidence of their impact on improving learning outcomes is inconsistent. The results of this study revealed, however, that wearing spectacles was significantly associated with improved academic performance among secondary schoolchildren with reduced VA and myopia in Hue City, Vietnam.

**Table 2. Characteristics of wearing spectacles among the study participants (n = 647).**

|  |  | Wearing spectacles |  |  |  | Total |
|---|---|---|---|---|---|---|
|  |  | Yes | % | No | % |  |
| **Reduced VA**[c] | Yes | 112 | 74.7 | 38 | 25.3 | 150 |
|  | No | 21 | 4.2 | 476 | 95.8 | 497 |
| **Refractive error** | Myopia | 116 | 36.8 | 199 | 63.2 | 315 |
|  | Emmetropia | 10 | 3.4 | 283 | 96.6 | 293 |
|  | Hyperopia | 7 | 17.9 | 32 | 82.1 | 39 |

[c]Reduced VA defined as VA ≤ 0.7 using the Landolt C chart

**Table 3. Distribution of reduced visual acuity (VA) and refractive error (n = 647).**

| Reduced VA[c] | n (%) | Refractor error | n (%) | % Total |
|---|---|---|---|---|
| Yes | 150 (23.2) | Myopia | 141 (94.0) | 21.8 |
|  |  | Emmetropia | 6 (4.0) | 0.9 |
|  |  | Hyperopia | 3 (2.0) | 0.5 |
| No | 497 (76.8) | Myopia | 174 (35.0) | 26.9 |
|  |  | Emmetropia | 287 (57.7) | 44.4 |
|  |  | Hyperopia | 36 (7.2) | 5.6 |

[c]Reduced VA defined as VA ≤ 0.7 using the Landolt C chart

**Table 4. Association between wearing spectacles and academic performance among students with reduced visual acuity (n = 150).**

| | GPA | Math score | Literature score |
|---|---|---|---|
| | β (95% CI) | β (95% CI) | β (95% CI) |
| **Model 1** | | | |
| Wearing spectacles | 0.470 (0.105–0.835)* | 0.534 (0.026–1.042)* | 0.441 (0.068–0.813)* |
| **Model 2** | | | |
| Sex (ref. female) | −0.341 (−0.658 to −0.024)* | 0.155 (−0.603 to 0.293) | −0.283 (−0.610 to 0.044) |
| Study time (ref. > 8 hours/day) | | | |
| ≤ 6 hours/day | −0.375 (−1.153 to 0.403) | −0.392 (−1.493 to 0.708) | 0.025 (−0.778 to 0.828) |
| 6–8 hours/day | −0.314 (−0.670 to 0.041) | −0.450 (−0.953 to 0.052) | −0.192 (−0.558 to 0.175) |
| Wearing spectacles | 0.448 (0.092–0.805)* | 0.502 (−0.002 to 1.005) | 0.430 (0.063–0.797)* |
| **Model 3** | | | |
| Sex (ref. male) | −0.381 (−0.699 to −0.062)* | −0.196 (−0.648 to 0.256) | −0.323 (−0.651 to 0.004) |
| Study time (ref. > 8 hours/day) | | | |
| ≤ 6 hours/day | −0.439 (−1.225 to 0.348) | −0.441 (−1.557 to 0.675) | −0.098 (−0.906 to 0.710) |
| 6–8 hours/day | −0.345 (−0.710 to 0.020) | −0.471 (−0.989 to 0.046) | −0.260 (−0.635 to 0.115) |
| Time spent on electronic devices for pleasure (ref. > 1 hours/day) | | | |
| Not at all | −0.752 (−2.110 to 0.606) | −0,859 (−2.787 to 1.068) | −0.481 (−1.877 to 0.914) |
| ≤ 1 hours/day | −0.984 (−2.383 to 0.415) | −1.053 (−3.039 to 0.932) | −0.873 (−2.311 to 0.565) |
| Wearing spectacles | 0.462 (0.108–0.816)* | 0.517 (0.015–1.020)* | 0.438 (0.074–0.802)* |

CI, confidence interval; GPA, grade point average

* p < 0.05

Myopia is considered a significant Asian public health issue, with its prevalence being notably higher in Asian than in Western countries [16–18], and is perceived to be a critical health problem in various urbanized areas (e.g., Singapore, Taipei, China, India) [19–22]. The results of this study showed that the prevalence of myopia among secondary school children was 48.7%. Similar to our study, autorefractometry has been performed under non-cycloplegic conditions in other studies, the results of which found that myopia is a common refractive error in schools in Vietnam [16,23]. Another study of rural secondary school children, however, showed a lower prevalence of students with myopia than what we observed [24]. The prevalence of myopia among Vietnamese schoolchildren is relatively higher than that in Western countries but similar to or lower than that in other Asian countries. Prior results from a study in China showed a slightly higher prevalence of myopia among primary and secondary school-aged children [25–28], whereas studies on urban and suburban Indian schoolchildren showed a lower prevalence [29,30]. These differences may be due to differences in subject age, cutoff thresholds for a diagnosis of myopia, or measurement methods (cycloplegic vs. non-cycloplegic). Additionally, the prevalence of reduced VA in both eyes was 23.3% in our study, slightly higher than that reported in other school-based studies conducted by the same group [31,32]. Ultimately, an accurate comparison of refractive error prevalence rates with other studies is difficult, owing to the use of different definitions, instrumentation, and variable use of cycloplegia, as well as differences in other population characteristics, such as ethnicity, sex, and age distribution.

**Table 5. Association between wearing spectacles and academic performance among students with myopia (n = 315).**

| | GPA<br>β (95% CI) | Math score<br>β (95% CI) | Literature score<br>β (95% CI) |
|---|---|---|---|
| **Model 1** | | | |
| Wearing spectacles | 0.237 (0.005–0.469)* | 0.252 (−0.070 to 0.574) | 0.319 (0.076–0.562)* |
| **Model 2** | | | |
| Sex (ref. female) | −0.340 (−0.555 to −0.125)** | −0.162 (−0.469 to 0.145) | −0.572 (−0.795 to −0.348)*** |
| Study time (ref. > 8 hours/day) | | | |
| ≤ 6 hours/day | −0.903 (−1.352 to −0.455)*** | −0.939 (−1.577 to −0.300)** | −0.591 (−1.056 to −0.126)* |
| 6–8 hours/day | −0.344 (−0.590 to −0.098)** | −0.436 (−0.787 to −0.086)* | −0.296 (−0.552 to −0.041)* |
| Wearing spectacles | 0.204 (−0.019 to 0.426) | 0.227 (−0.090 to 0.544) | 0.280 (0.049 to 0.511)* |
| **Model 3** | | | |
| Sex (ref. male) | −0.336 (−0.555 to −0.121)** | −0.166 (−0.472 to 0.141) | −0.567 (−0.791 to −0.344)*** |
| Study time (ref. > 8 hours/day) | | | |
| ≤ 6 hours/day | −0.908 (−1.356 to −0.460)*** | −0.950 (−1.588 to −0.312)** | −0.590 (−1.055 to −0.125)* |
| 6–8 hours/day | −0.352 (−0.599 to −0.106)** | −0.435 (−0.786 to −0.084)* | −0.305 (−0.561 to −0.049)* |
| Time spent on electronic devices for pleasure (ref. > 1 hours/day) | | | |
| Not at all | 0.152 (−0.229 to 0.532) | −0.055 (−0.597 to 0.487) | 0.158 (−0.237 to 0.554) |
| ≤ 1 hours/day | 0.123 (−0.115 to 0.362) | 0.192 (−0.148 to 0.532) | 0.023 (−0.225 to 0.271) |
| Wearing spectacles | 0.194 (−0.029 to 0.417) | 0.212 (−0.105 to 0.530) | 0.277 (0.046–0.509)* |

CI, confidence interval; GPA, grade point average

* p < 0.05; ** p < 0.01; *** p < 0.001

The study also found that nearly two-thirds (63.2%) of students with myopia and a quarter (25.3%) of students with reduced VA did not wear spectacles during the assessment. Although multiple studies have emphasized that wearing spectacles has a definitive impact on slowing the progression of myopia [33–36], a significant number of students with myopia do not wear spectacles. A similarly high and unmet need for refractive correction has been documented in previous studies. A prior study in Vietnam indicated that about two of three secondary school students with refractive errors have not received the appropriate correction [37]. Among other populations, results from India showed that 75% of children with myopia did not wear spectacles [38], while a study from China showed that wearing spectacles was observed in only 37% of children with visual impairments. Despite better access to eye doctors and optical shops in urban areas, there are still urban schoolchildren with untreated refractive errors. A large number of myopic students do not wear spectacles in Vietnam. This situation in Vietnam and other low- and middle-income countries could be attributed to reasons that include a lack of comprehensive vision health education, particularly regarding the benefits of spectacles in treating refractive errors and insufficient financial support for vision health promotion. Furthermore, existing evidence indicates that children do not like spectacles or are self-conscious of their appearance in them [39]; therefore, having children comply with wearing spectacles should be a concern, indicating a need for health education regarding wearing spectacles and vision screening, as research has shown that eye health promotion interventions significantly improve eye health knowledge, attitudes, and practices of school children [40].

The main finding of this study was the significant association between spectacle use and academic performance (e.g., GPA, math, and literature scores) among schoolchildren with reduced VA and myopia. This relationship remained significant after adjusting for sex, study time, and recreational use of electronic devices for the reduced VA group (model 3). In the myopia group, wearing spectacles was significantly associated with higher Vietnamese literature scores but not with GPA or math scores, after controlling for sex, study time, and time spent on electronic devices for leisure. Students with

myopia who wore spectacles had significantly higher scores in Vietnamese literature than their counterparts. In addition, associations between spectacle use and academic achievements were stronger among the students with reduced VA (Table 4) than among students with myopia (Table 5). The findings from our study corroborated those of previous studies in different populations. A study from China revealed that wearing spectacles improved the Chinese and math test scores of students with myopia by 0.27 and 0.24 standard deviations, respectively [18], while another study showed that better VA, but not cycloplegia refractive error or wearing spectacles, was significantly associated with academic performance [41]. This difference might be because academic performance was measured by standardized examinations that included five subjects, instead of measuring each subject as in our study. Other studies, including a cluster randomized clinical trial in the United States, have shown that students who wore spectacles achieved better reading scores than those who did not [42]. While this study focused on the link between spectacle-wearing and academic performance, other research has also found a positive correlation between global education levels and myopia prevalence [43]. Evidence from China suggests that refractive errors are linked to academic performance, as students with better academic performance tend to have faster myopia development [44]. The association between wearing spectacles and academic performance can be explained easily, as being able to see clearly at a distance is crucial for students to learn, especially since most secondary schools often use blackboards or whiteboards as the main tool of instruction, along with digital displays.

The strength of our study lies in the availability of different measures of academic performance, including GPAs, math scores, and literature scores. This study had a few limitations. First, it should be noted that autorefractometry in this study was not performed under cycloplegic conditions. Non-cycloplegic autorefractometry can overestimate the prevalence of myopia in adolescents owing to active accommodation, but it is commonly used because it is quicker and more comfortable for students, making it particularly useful in school vision screenings [45,46]. Thus, future research should incorporate cycloplegic conditions to improve the accuracy and reliability of myopia assessments. Second, independent variables such as study time and time spent on electronic devices for pleasure were collected through face-to-face interviews and might have been subject to recall bias. Despite these limitations, this study provides suggestive evidence that improved compliance in wearing spectacles leads to enhanced academic outcomes in secondary school children. The current findings provide an additional impetus for concerted efforts from government ministries, schools, parents, and healthcare professionals to ensure that no child is denied access to good education because of poor vision.

## Supporting information

**S1 File. Dataset used for the analysis.**
(CSV)

## Acknowledgments

The authors would like to thank Prof. Nguyen Vu Quoc Huy for overarching advice on the research and support of data collection by the teams of the Faculty of Public Health and Department of Ophthalmology of Hue University of Medicine and Pharmacy. We would also like to thank the Department of Education and Training in Thua Thien Hue Province, Vietnam, for granting permission for their schools to participate in this study, the schools and teachers who kindly agreed to administer the questionnaire, and all of the students who generously agreed to participate.

## Author contributions

**Conceptualization:** Hoang Thuy Linh Nguyen, Keiko Nakamura, Kaoruko Seino, Kyoko Ohno-Matsui.

**Data curation:** Hoang Thuy Linh Nguyen, Keiko Nakamura.

**Formal analysis:** Hoang Thuy Linh Nguyen, Xuan Minh Tri Tran, Keiko Nakamura.

**Funding acquisition:** Keiko Nakamura.

**Investigation:** Hoang Thuy Linh Nguyen, Keiko Nakamura.

**Methodology:** Hoang Thuy Linh Nguyen, Keiko Nakamura, Kaoruko Seino.

**Project administration:** Hoang Thuy Linh Nguyen, Keiko Nakamura.

**Resources:** Hoang Thuy Linh Nguyen, Keiko Nakamura.

**Supervision:** Keiko Nakamura, Van Thang Vo, Kyoko Ohno-Matsui.

**Validation:** Hoang Thuy Linh Nguyen, Yuri Tashiro, Ayano Miyashita, Tae Igarashi-Yokoi, Van Thang Vo, Kyoko Ohno-Matsui.

**Writing – original draft:** Hoang Thuy Linh Nguyen, Xuan Minh Tri Tran.

**Writing – review & editing:** Hoang Thuy Linh Nguyen, Xuan Minh Tri Tran, Keiko Nakamura, Kaoruko Seino, Yuri Tashiro, Ayano Miyashita, Tae Igarashi-Yokoi, Van Thang Vo, Kyoko Ohno-Matsui.

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
