## [Decision Letter · Decision Letter 0]

15 Oct 2024

PONE-D-24-39987The impact of eyeglasses on academic performance among Vietnamese adolescents with low vision and myopia: a school-based studyPLOS ONE

Dear Dr. Nakamura,

Thank you for submitting your manuscript to PLOS ONE. After careful consideration, we feel that it has merit but does not fully meet PLOS ONE’s publication criteria as it currently stands. Therefore, we invite you to submit a revised version of the manuscript that addresses the points raised during the review process.

**ACADEMIC EDITOR:**

Dear Authors,

Thanks you for submitting your paper to Plos One.

We have now completed the review process and happy to share with you reviewers comments.

Please address comments from both reviewers and submit a revised manuscript.

Thanks.

We look forward to receiving your revised manuscript.

Kind regards,

Baskar Theagarayan, Ph.D

Academic Editor

PLOS ONE

Journal Requirements:

“Authors declare that the research was conducted in the absence of any commercial or financial relationships that could be construed as a potential conflict of interest.”

Reviewers' comments:

Reviewer's Responses to Questions

**Comments to the Author**

1. Is the manuscript technically sound, and do the data support the conclusions?

Reviewer #1: Yes

Reviewer #2: Partly

2. Has the statistical analysis been performed appropriately and rigorously? 

Reviewer #1: Yes

Reviewer #2: I Don't Know

3. Have the authors made all data underlying the findings in their manuscript fully available?

Reviewer #1: Yes

Reviewer #2: No

4. Is the manuscript presented in an intelligible fashion and written in standard English?

Reviewer #1: Yes

Reviewer #2: Yes

5. Review Comments to the Author

Reviewer #1: The manuscript entitled “The impact of eyeglasses on academic performance among Vietnamese adolescents with low vision and myopia: a school-based study” provides a thorough analysis of the relationship between myopia, wearing eyeglasses, and academic performance. This study, conducted through a rigorous set of eye examinations and structured questionnaires, emphasizes the advantages of wearing glasses for students with low vision and myopia to enhance their academic performance. However, there are several concerns that require attention:

Major Concerns:

1. The correlation between myopia and academic performance may be tricky. Were students with better academic performance more likely to develop myopia? Understanding this aspect is crucial for interpreting the study's findings.

2. It is essential to clarify in this cross-sectional study whether the students' eyeglasses completely corrected their refractive errors. Were the academic scores collected before the eye examinations retrospectively? Clarity on the vision improvement due to wearing glasses is necessary to comprehend the impact of eyeglasses.

3. Rather than solely focusing on how eyeglasses enhance academic performance, exploring how myopia affects academic achievements and identifying cases where eyeglasses are essential would be beneficial. The authors could delve into their data to determine instances where students experienced a significant decline in scores due to vision loss, demonstrating the potential benefits of wearing glasses for such students

4. Additionally, It would be interesting to investigate whether myopia was linked to study duration or study time based on the available data.

Minor Concerns: There are some minor writing issues in the manuscript. For instance, in the abstract's conclusion, the study cannot definitively state that myopia represents a significant public health issue. Additionally, a comma is missing after "do not" in the sentence, "Although those students are expected to wear proper glasses to correct refractive errors, a large proportion of them do not which is expected to lead to their vision problems."

Overall, this manuscript is well-written and offers a valuable contribution to the field. The study's outcomes carry substantial clinical relevance and practical implications.

Reviewer #2: There are some usages of language that needs to be updated: glasses needs to be updated to spectacles or spectacle wear; sex needs to be updated to gender.

There is a lack of clarity in the sample groupings that needs to be explained more clearly, in order to interpret the analysis.

For example: there is a definition of low vision, where the VA is <7/10, which is likely presenting vision, as UVA is utilised. These participants are grouped against another set of participants who also have myopia and wore glasses (n=116/315), however it is unclear how or why they were grouped in the analysis, as the majority of participants who were termed low vision did have a pair of spectacles (n=112/150). What was the reason for two sets of myopes, with spectacles to be separated in this way? Was there a record of the VA with their spectacles, for both sets of participants?

The whole study was done on 11yo children with no cycloplegic and it seems no actual check of the refraction, only an autorefraction, which does pose a problem with reliability.

A more detailed labelling of table 4 would be helpful, as an interpretation of the association shows "wearing glasses" in both the low vision group and within that group in the two models: does this mean that "wearing glasses" was utilised as both an independent and dependent variable? The association of a variable with itself, in either model, seems to be problematic.

6. PLOS authors have the option to publish the peer review history of their article (what does this mean? ). If published, this will include your full peer review and any attached files.

**Do you want your identity to be public for this peer review?** For information about this choice, including consent withdrawal, please see our Privacy Policy .

Reviewer #1: No

Reviewer #2: No

---

## [Author Response · Author response to Decision Letter 1]

6 Dec 2024

Reviewer #1: The manuscript entitled “The impact of eyeglasses on academic performance among Vietnamese adolescents with low vision and myopia: a school-based study” provides a thorough analysis of the relationship between myopia, wearing eyeglasses, and academic performance. This study, conducted through a rigorous set of eye examinations and structured questionnaires, emphasizes the advantages of wearing glasses for students with low vision and myopia to enhance their academic performance. However, there are several concerns that require attention:

Major Concerns:

1. The correlation between myopia and academic performance may be tricky. Were students with better academic performance more likely to develop myopia? Understanding this aspect is crucial for interpreting the study's findings.

>>>Response:

According to the reviewer’s comment, the authors carefully address the relationship between academic performance and myopia. The relationship between myopia and academic performance could be bidirectional and this relationship was discussed based on literature (Page 11, Lines 224-228). A similar correlation was found in our data between GPA (r=0.167, p<0.001), math score (r=0.130, p<0.001), literature score (r=0.174, p<0.001) and myopia. Our study focused on the relationship between wearing eyeglasses and academic performance among myopia students.

2. It is essential to clarify in this cross-sectional study whether the students' eyeglasses completely corrected their refractive errors. Were the academic scores collected before the eye examinations retrospectively? Clarity on the vision improvement due to wearing glasses is necessary to comprehend the impact of eyeglasses.

>>>Response:

According to the reviewer’s comment, the authors added a sentence addressing wearing classes as a solution for correcting refractive errors (Page 3, Lines 62-64), but students’ eyeglasses might not completely correct their refractive errors (Page 3, Lines 65-66). The academic scores were collected at the end of the school term, independent of the eye examinations. Our study focused on the relationship between wearing eyeglasses and academic performance among myopia students, but not on a causal relationship.

3. Rather than solely focusing on how eyeglasses enhance academic performance, exploring how myopia affects academic achievements and identifying cases where eyeglasses are essential would be beneficial. The authors could delve into their data to determine instances where students experienced a significant decline in scores due to vision loss, demonstrating the potential benefits of wearing glasses for such students.

>>>Response:

The authors appreciate the reviewer’s suggestion to explore an important issue. Due to the cross-sectional nature of the study, this question could be answered by our future research. While leaving an answer to future studies, we have conducted an additional analysis based on the current data and found that students with myopia tended to have better academic performance compared to those without myopia. Furthermore, the present analysis showed that within the group of students with myopia, those who wore glasses had higher literature scores than those who did not, emphasizing the benefits of wearing glasses on academic performance.

4. Additionally, It would be interesting to investigate whether myopia was linked to study duration or study time based on the available data.

>>>Response:

In response to the reviewer’s comment, we performed additional analysis and found that students who studied for less than 6 hours were less likely to develop myopia compared to those who studied for more than 8 hours (OR=0.40; 95% CI 0.21, 0.77; p<0.01).

Minor Concerns: There are some minor writing issues in the manuscript. For instance, in the abstract's conclusion, the study cannot definitively state that myopia represents a significant public health issue. Additionally, a comma is missing after "do not" in the sentence, "Although those students are expected to wear proper glasses to correct refractive errors, a large proportion of them do not which is expected to lead to their vision problems."

>>>Response:

According to the reviewer’s comments, the sentences were amended (Page 2, Lines 39-41).

Overall, this manuscript is well-written and offers a valuable contribution to the field. The study's outcomes carry substantial clinical relevance and practical implications.

----------

Reviewer #2: There are some usages of language that needs to be updated: glasses needs to be updated to spectacles or spectacle wear; sex needs to be updated to gender.

>>>Response:

According to the reviewer’s comments, “glasses” were replaced with “spectacle use”. In this paper, we use “sex” to represent the biological characteristics of males and females, rather than the socially constructed characteristics of women and men.

There is a lack of clarity in the sample groupings that needs to be explained more clearly, in order to interpret the analysis.

For example: there is a definition of low vision, where the VA is <7/10, which is likely presenting vision, as UVA is utilised. These participants are grouped against another set of participants who also have myopia and wore glasses (n=116/315), however it is unclear how or why they were grouped in the analysis, as the majority of participants who were termed low vision did have a pair of spectacles (n=112/150). What was the reason for two sets of myopes, with spectacles to be separated in this way? Was there a record of the VA with their spectacles, for both sets of participants?

>>>Response:

According to the reviewer’s comment, the authors clarified the measurement was uncorrected visual acuity and amended the term “low vision” to “reduced visual acuity (VA)”. Categorizing students with reduced VA and myopia was made independently. Therefore, Table 4 was divided into new Tables 4 (analysis with students with reduced VA) and new Table 5 (analysis with students with myopia). Amendments were made to clarify the sample groupings and model descriptions for Tables 4 and 5 (Page 5, Line 119 – Page 6, Line 124), and the heading of the tables (Page 8, Lines 158-159; Page 8, Lines 166-167). In these tables, the variable for wearing glasses was included solely as an independent variable across the three models.

The whole study was done on 11yo children with no cycloplegic and it seems no actual check of the refraction, only an autorefraction, which does pose a problem with reliability.

>>>Response:

According to the reviewer’s comments, the authors discussed a limitation of the use of noncycloplegic autorefraction in this study (Page 11, Lines 233-237).

A more detailed labelling of table 4 would be helpful, as an interpretation of the association shows "wearing glasses" in both the low vision group and within that group in the two models: does this mean that "wearing glasses" was utilised as both an independent and dependent variable? The association of a variable with itself, in either model, seems to be problematic.

>>>Response:

According to the reviewer’s comment, the former Table 4 was divided into Tables 4 (analysis with students with reduced visual acuity) and 5 (analysis with students with myopia). Amendments were made to clarify the sample groupings and model descriptions for Tables 4 and 5 (Page 5, Line 119 – Page 6, Line 124), and the heading of the tables (Page 8, Lines 158-159; Page 8, Lines 166-167). In these tables, the variable for wearing glasses was included solely as an independent variable across the three models.

<responses to the detailed comments from Reviewer 2 on the pdf file>

Consider SPECS 2030 initiative as a relevant point in the need for the study

>>>Response:

According to the reviewer’s comment, the sentences were amended (Page 2, Lines 22-23; Page 3, Lines 59-60).

Consider spectacles instead of glasses throughout the text

>>>Response:

According to the reviewer’s comment, “glasses” was replaced with “spectacles” throughout the manuscript.

They have a refractive error, what does this sentence mean "lead to their vision problems"?

>>>Response:

According to the reviewer’s comment, the sentence was amended (Page 2, Lines 39-40).

How has this been proven to be the only cause in these cases?

>>>Response:

According to the reviewer’s comment, the sentence was amended (Page 3, Lines 54-55).

However, this is not a "fix all" solution for myopes who are known to progress.

>>>Response:

According to the reviewer’s comment, the sentence was amended (Page 3, Lines 62-64).

Provide calculation that this sample size is representative of the Province total; provide how the population of 70000 was determined

>>>Response:

According to the reviewer’s comment, a paragraph addressing study population was amended (Page 4, Lines 77-78). The total number of secondary-school students and multistage stratified cluster random sampling procedure were elaborated.

How reliable is this in the particular age group of children?

>>>Response:

According to the reviewer’s comment, a sentence was added to address the limitation (Page 11, Lines 233-237).

Has the questionnaire undergone a validation through statistical analysis in addition to face validity?

>>>Response:

According to the reviewer’s comments, a paragraph was amended (Page 5, Lines 96-103).

This is not an accepted definition of low vision, you will have to adapt this. this is potentially just uncorrected and not a true reflection of low vision, "presenting vision".

>>>Response:

According to the reviewer’s comments, “low vision” was replaced with “reduced visual acuity” throughout the manuscript.

If they were classified as low vision, does this mean that they owned a pair of glasses but did not use them at the time of assessment? This is crucial to establish if children were wearing their glasses some of the time as this will impact their learning and thus reduce the reliability of the findings if this was not taken into account.

>>>Response:

According to the reviewer’s comment, the sentences to elaborate “reduced visual acuity” (Page 5, Lines 105-107) and spectacle use (Page 5, Lines 111-112) were amended. This classification of spectacle use did not account for whether the students owned spectacles but focused on their actual use at the time of examination.

Why were they classified as low vision? They had specs but were not present at the assessment? they were ineffective? they were used in the assessment but the VA was still poor?

>>>Response:

According to the reviewer’s comments, “reduced visual acuity” was used instead of “low vision” throughout the manuscript. The definition was elaborated (Page 5, Lines 105-107).

Myopia cannot increase their score; this is not well described. Students with myopia who wore spectacles had a higher GPA score; when compared to students without myopia or need of a refractive correction.

>>>Response:

According to the reviewer’s comment, a sentence was modified (Page 8, Lines 161-162).

Children with reduced vision now in spectacles? how is this parameter different from "wearing spectacles" in model 1 and model 2?

>>>Response:

According to the reviewer’s comment, the former Table 4 was divided into Tables 4 (analysis with students with reduced visual acuity) and 5 (analysis with students with myopia). In both tables, the variable for wearing glasses was included solely as an independent variable across the three models.

Please clarify the methodology and analysis; wearing glasses cannot be both an independent and dependent variable, which is what I understand as being depicted in this table; these would create high associations, which is not a reliable analysis.

>>>Response:

According to the reviewer’s comment, a paragraph in the Material and methods section was amended to clarify the sample groupings and model descriptions for Tables 4 and 5 (Page 5, Line 119 – Page 6, Line 124). In these tables, the variable for wearing glasses was included solely as an independent variable across the three models.

Then the majority of these school children that were categorised as low vision did have a pair of spectacles? Why were they in this category if they had a correction? please clarify

>>>Response:

According to the reviewer’s comment, a sentence was modified (Page 10, Lines 194-195).

This is perhaps misleading: your results show an association; this is not cause and "effect" per se of the physical glasses rather as a response to the visual outcome of glasses i.e: they can see better; so it is easier to do better

>>>Response:

According to the reviewer’s comment, the sentence was amended (Page 11, Lines 211-212).

---

## [Decision Letter · Decision Letter 1]

14 Jan 2025

PONE-D-24-39987R1Impact of spectacle use on academic performance among Vietnamese adolescents with reduced visual acuity and myopia: A school-based studyPLOS ONE

Dear Dr. Nakamura,

Thank you for submitting your manuscript to PLOS ONE. After careful consideration, we feel that it has merit but does not fully meet PLOS ONE’s publication criteria as it currently stands. Therefore, we invite you to submit a revised version of the manuscript that addresses the points raised during the review process.

We look forward to receiving your revised manuscript.

Kind regards,

Baskar Theagarayan, Ph.D

Academic Editor

PLOS ONE

Journal Requirements:

Additional Editor Comments:

Dear Authors,

Thanks for submitting your revised version and answering both reviewers comments.

One of the reviewers have made small further comments, please address these and also proof read and submit so we accept it for publication.

Regards,

Baskar

Reviewers' comments:

Reviewer's Responses to Questions

**Comments to the Author**

1. If the authors have adequately addressed your comments raised in a previous round of review and you feel that this manuscript is now acceptable for publication, you may indicate that here to bypass the “Comments to the Author” section, enter your conflict of interest statement in the “Confidential to Editor” section, and submit your "Accept" recommendation.

Reviewer #1: (No Response)

Reviewer #2: All comments have been addressed

2. Is the manuscript technically sound, and do the data support the conclusions?

Reviewer #1: Yes

Reviewer #2: Yes

3. Has the statistical analysis been performed appropriately and rigorously? 

Reviewer #1: Yes

Reviewer #2: Yes

4. Have the authors made all data underlying the findings in their manuscript fully available?

Reviewer #1: Yes

Reviewer #2: Yes

5. Is the manuscript presented in an intelligible fashion and written in standard English?

Reviewer #1: Yes

Reviewer #2: Yes

6. Review Comments to the Author

Reviewer #1: The authors have provided thorough and thoughtful responses to the reviewers' comments. Sentence amendment and word replacement were updated, and the explanation for retaining "sex" is concise and justified. Additional analysis on study time was performed and separate tables were made to clarify the sample grouping. The explanations regarding "wearing glasses" as an independent variable also address the reviewer’s methodological concerns. The authors have commendably acknowledged the limitations of the study's cross-sectional nature while performing an additional analysis to investigate performance differences among groups. The explanation regarding future research is reasonable but could be expanded to suggest specific longitudinal or interventional study designs that would help address this concern.

Below are detailed evaluations and suggestions for further refinement:

Major Concerns:

1. Separate tables were made to clarify the sample grouping. But there is a lack of exploration into the possibility of differences in outcomes between the VA group and the myopia group.

2. For the 35% of the myopia group without reduced visual acuity (VA), the importance of wearing glasses needs to be demonstrated.

Minor Concerns:

1. In Page 3, Lines 65-66, the authors updated the global data that “only 36% of people with distance vision impairment due to refractive errors have access to appropriate spectacles” The citation needs to be added.

Reviewer #2: Thank you for the resubmission; all changes have been addressed. I have found this to be a very interesting paper and it offers valuable insight into the broad topic of myopia progression in children.

7. PLOS authors have the option to publish the peer review history of their article (what does this mean? ). If published, this will include your full peer review and any attached files.

**Do you want your identity to be public for this peer review?** For information about this choice, including consent withdrawal, please see our Privacy Policy .

Reviewer #1: No

Reviewer #2: No

---

## [Author Response · Author response to Decision Letter 2]

16 Feb 2025

Major Concerns:

1. Separate tables were made to clarify the sample grouping. But there is a lack of exploration into the possibility of differences in outcomes between the VA group and the myopia group.

--- Response:

Thank you for expressing this concern. The distributions of VA group and myopia group differ, as shown in Table 3. We have revised the sentence in the Discussion section clearly stating that the associations between spectacle use and academic achievements were stronger among the students with reduced VA (Table 4) than among students with myopia (Table 5).” (Page 11, Lines 217-219)

2. For the 35% of the myopia group without reduced visual acuity (VA), the importance of wearing glasses needs to be demonstrated.

--- Response:

In response to the reviewer’s comments, there are several reasons why students with myopia may not experience noticeable visual impairment. One factor is the accommodative mechanism and the use of non-cycloplegic measurements, which we have already acknowledged in the limitations section (Page 11, Lines 234-238). Furthermore, we have tested the association between wearing spectacles and academic performance among students with myopia but no reduced VA (n=174). Results showed that wearing spectacles was significantly associated with improved academic performance among students with myopia but reduced VA, with increased math scores (β = 1.108; P = 0.023) compared with those of students not wearing spectacles, remained significantly associated after adjusting for sex, study time, and time spent on electronic devices for pleasure. Thus, use of spectacle is beneficial to students with myopia even without reduced VA.

Minor Concerns:

1. In Page 3, Lines 65-66, the authors updated the global data that “only 36% of people with distance vision impairment due to refractive errors have access to appropriate spectacles” The citation needs to be added.

--- Response:

In response to the reviewer’s comments, the sentence was cited (ref. 13, in Page 3, Lines 64-65)

---

## [Decision Letter · Decision Letter 2]

25 Mar 2025

Impact of spectacle use on academic performance among Vietnamese adolescents with reduced visual acuity and myopia: A school-based study

PONE-D-24-39987R2

Dear Dr. Nakamura,

We’re pleased to inform you that your manuscript has been judged scientifically suitable for publication and will be formally accepted for publication once it meets all outstanding technical requirements.

Kind regards,

Baskar Theagarayan, Ph.D

Academic Editor

PLOS ONE

Additional Editor Comments (optional):

Reviewers' comments:

Reviewer's Responses to Questions

**Comments to the Author**

1. If the authors have adequately addressed your comments raised in a previous round of review and you feel that this manuscript is now acceptable for publication, you may indicate that here to bypass the “Comments to the Author” section, enter your conflict of interest statement in the “Confidential to Editor” section, and submit your "Accept" recommendation.

Reviewer #1: All comments have been addressed

2. Is the manuscript technically sound, and do the data support the conclusions?

Reviewer #1: Yes

3. Has the statistical analysis been performed appropriately and rigorously? 

Reviewer #1: Yes

4. Have the authors made all data underlying the findings in their manuscript fully available?

Reviewer #1: Yes

5. Is the manuscript presented in an intelligible fashion and written in standard English?

Reviewer #1: Yes

6. Review Comments to the Author

Reviewer #1: The authors have addresses all the comments well. It should be ready to be published.

7. PLOS authors have the option to publish the peer review history of their article (what does this mean? ). If published, this will include your full peer review and any attached files.

**Do you want your identity to be public for this peer review?** For information about this choice, including consent withdrawal, please see our Privacy Policy .

Reviewer #1: No

---

## [Editor Report · Acceptance letter]

PONE-D-24-39987R2

PLOS ONE

Dear Dr. Nakamura,

I'm pleased to inform you that your manuscript has been deemed suitable for publication in PLOS ONE. Congratulations! Your manuscript is now being handed over to our production team.

Kind regards,

on behalf of

Dr Baskar Theagarayan

Academic Editor

PLOS ONE